# The Occlusal Contact Area Is Associated with the Magnitude but Not Peak Timing of Changes in Chewing-Induced Brain Blood Flow in Mandibular Prognathism

**DOI:** 10.3390/dj13060250

**Published:** 2025-06-03

**Authors:** Hiroyuki Kanzaki, Chihiro Kariya, Kana Yoshida, Yuri Inagawa, Masao Kumazawa, Hiroshi Tomonari

**Affiliations:** Department of Orthodontics, School of Dental Medicine, Tsurumi University, Yokohama 230-8501, Japantomonari-h@tsurumi-u.ac.jp (H.T.)

**Keywords:** brain blood flow, malocclusion, food hardness, jaw deformity, mandibular prognathism

## Abstract

**Background/Objectives**: Jaw deformities affect not only facial aesthetics but also various oral functions. While previous studies have demonstrated that mandibular prognathism (MP) alters masticatory-induced brain blood flow (BBF), the temporal characteristics of these hemodynamic changes have remained unclear. In this cross-sectional observational study, we investigated the following two specific objectives: (1) whether food hardness affects not only the magnitude but also the temporal patterns of BBF changes during mastication and (2) how malocclusion is associated with these temporal hemodynamic responses. **Methods**: Twenty-six participants with normal occlusion (NORM) and twenty patients with MP participated in this study. BBF was measured using functional near-infrared spectroscopy, while participants chewed soft paraffin or hard gummy candy. Maximum oxygenated hemoglobin (oxy-Hb) values and time-to-peak BBF were analyzed. **Results**: While food hardness did not significantly affect maximum oxy-Hb within groups, the MP group showed significantly lower responses during hard gummy candy mastication compared to the NORM group. The occlusal contact area exhibited significant positive correlation with maximum oxy-Hb values, while the ANB angle, an indicator of intermaxillary skeletal relationship, showed no significant correlation with BBF parameters. The hard gummy candy/paraffin ratio of maximum oxy-Hb was significantly higher in the NORM group compared to the MP group. Time-to-peak BBF was approximately twice as long for hard gummy candy compared to paraffin in both groups, with no significant differences between groups. **Conclusions**: These findings reveal that while MP attenuates the magnitude of masticatory-induced BBF, particularly during hard food mastication, the temporal adaptation to increased food hardness is preserved. This dissociation between magnitude and timing effects suggests that intact basic neurovascular coupling mechanisms would be maintained even in the condition of altered masticatory function in a MP subject, which is providing new insights for rehabilitation strategies in orthognathic surgery cases.

## 1. Introduction

Malocclusion, such as jaw deformities, not only affects facial aesthetics but also significantly affects various oral and maxillofacial functions, including mastication, speech, and breathing [1,2,3]. Patients with jaw deformities—skeletal mandibular prognathism (MP) and skeletal open bite—exhibit altered brain blood flow (BBF) patterns during mastication compared to individuals with normal occlusion [4,5], suggesting a potential relationship between occlusal condition and brain hemodynamics during masticatory function.

The relationship between mastication and BBF has been previously extensively documented. A recent systematic review demonstrated that mastication consistently activates multiple brain regions and enhances regional BBF, particularly in the sensorimotor cortex and prefrontal areas [6]. This masticatory-induced enhancement of BBF has been consistently observed across different imaging modalities, including functional magnetic resonance imaging (fMRI) [7] and near-infrared spectroscopy (NIRS) [8]. The observed increases in BBF during mastication reflect the complex neuromuscular coordination required for proper masticatory function, involving both sensory processing and motor control [9,10]. Furthermore, this mastication-induced BBF increase may have potential cognitive benefits, particularly in maintaining cognitive function in aging populations [11]. Collectively, these findings underscore the significant impact of masticatory function on brain hemodynamics.

The hardness of food is known to affect masticatory muscle activity and BBF during mastication [12,13,14]. Previous studies using fMRI have reported that moderately hard food elicits greater BBF responses in the sensorimotor cortex compared to soft food [15]. Notably, Hasegawa et al. demonstrated that while mastication of moderately hard gum increased BBF, excessively hard materials led to a decrease in BBF [16]. However, the detailed mechanisms underlying these phenomena and the optimal range of food hardness for maintaining adequate BBF during mastication remain to be fully elucidated.

Various neuroimaging techniques have been employed to investigate BBF during mastication. fMRI has been widely used due to its superior spatial resolution, allowing for precise localization of brain activity patterns [12,17]. fMRI studies have successfully mapped the activation of specific brain regions during mastication, particularly in the sensorimotor cortex and cerebellum [15]. However, NIRS offers distinct advantages in temporal resolution compared to fMRI, enabling real-time monitoring of hemodynamic changes [18,19]. Despite these methodological capabilities, few studies have focused on the temporal dynamics of BBF changes during mastication, and the temporal characteristics, including time-to-onset response, peak timing, and recovery patterns, remain largely unexplored.

While numerous studies have investigated the relationship between mastication and brain blood flow, these investigations have predominantly focused on the magnitude of hemodynamic changes, neglecting the temporal dynamics of these responses. Hasegawa et al. [16] demonstrated that food hardness affects the magnitude of masticatory-induced BBF, but the temporal characteristics of these responses remain largely unexplored. This represents a significant gap in our understanding, as temporal hemodynamic patterns provide crucial insights into neurovascular coupling mechanisms that magnitude measurements alone cannot capture [19,20].

Current understanding of masticatory-induced cerebrovascular dynamics remains incomplete in three critical dimensions:While existing studies have documented the spatial distribution of neural activation during mastication [12,15], the temporal evolution of these hemodynamic responses remains poorly characterized. The few studies exploring temporal aspects have been limited by methodological constraints, with fMRI offering superior spatial resolution but insufficient temporal resolution to capture rapid hemodynamic transitions [17,18].The interaction between food hardness and temporal BBF dynamics represents an unexplored area with significant functional implications. While Hasegawa et al. [16] demonstrated that food hardness affects the magnitude of masticatory-induced BBF, the temporal characteristics of these responses remain largely unexplored.How craniofacial malformations, such as mandibular prognathism, might alter these temporal patterns—independent of their known effects on BBF magnitude [4,5]—remains unknown. This temporal dimension has direct clinical relevance, as it may reveal compensatory neurovascular adaptations in patients with craniofacial deformities [21,22].

The present study aimed to investigate the temporal characteristics of masticatory-induced BBF changes using NIRS. The two specific objectives were to examine (1) whether food hardness affects not only the magnitude but also the temporal patterns of BBF changes during mastication and (2) how malocclusion is associated with these temporal hemodynamic responses. We hypothesized that both food hardness and occlusal condition would significantly influence the temporal dynamics of BBF, specifically in terms of response onset, peak timing, and recovery patterns.

## 2. Materials and Methods

### 2.1. Ethical Approval

This study was approved by the ethics committee of Tsurumi University School of Dental Medicine (approval numbers: 1316 and 121005) and conformed to the principles of the Declaration of Helsinki. Written informed consent was obtained from all participants before study commencement. The study followed STROBE guidelines for human observational studies.

### 2.2. Sample Size Calculation

Prior to the experiment, we estimated the required sample size using the G*Power 3.1.9.4 program (Universität Kiel, Kiel, Germany), with the following parameter settings: effect size = 0.8, alpha (significance level) = 0.05, power = 0.8, and allocation ratio of the groups = 2. The minimum total sample size was computed as 46, and our total sample number (46) was the same as the estimated number. Post hoc power analysis was also conducted using G*Power.

### 2.3. Participants

The study included two groups: participants with normal occlusion (NORM group, *n* = 26) comprising 7 men and 19 women, and patients with MP (MP group, *n* = 20), comprising 4 men and 16 women.

The NORM group met the following inclusion criteria:No defects other than missing third molars;Adequate overjet and overbite (2–4 mm each);Skeletal and dental midline deviation < 1.0 mm;No temporomandibular joint (TMJ) disorders diagnosed using the standardized Diagnostic Criteria for Temporomandibular Disorders (DC-TMD) protocol [23]. The DC-TMD examination included assessment of pain during mandibular movements, range of motion measurements, joint sounds, muscle and TMJ palpation, and functional limitations. Participants were excluded if they exhibited any signs or symptoms of TMJ disorders according to the DC-TMD.Angle Class I molar relationship;A normal intermaxillary relationship as judged by experienced orthodontists.

The MP group were patients who visited our hospital for the first time for surgical orthodontic treatment. They met the following inclusion criteria:No congenital abnormalities;No missing teeth other than third molars;Anterior crossbite;Skeletal and dental midline deviation of 1.0 mm or less from facial midline;No functional symptoms, such as TMJ disorders;No history of orthodontic treatment;Angle Class III molar relationship where the mandibular first molar exhibits an anterior position compared to the maxillary first molar;A skeletal Class III intermaxillary relationship in ANB angle.

The patients with systemic diseases or under medication use that could affect cerebrovascular regulation or neuromuscular function were excluded from the research subjects. Specifically, we excluded participants with the following:Cardiovascular disorders (hypertension, arrhythmia, and history of stroke);Neurological conditions that could affect sensorimotor function or cerebral blood flow (epilepsy, migraine, and traumatic brain injury);Psychiatric disorders;Metabolic disorders (diabetes mellitus, thyroid dysfunction);Current use of medications with known effects on cerebral blood flow or neuromuscular function (including beta-blockers, calcium channel blockers, anticonvulsants, antidepressants, muscle relaxants, and vasodilators);Regular use of substances that could affect central or peripheral vascular tone.

All participants completed a comprehensive health questionnaire and underwent a detailed interview to ensure they met these criteria.

### 2.4. Measurement of Occlusal Contact Area

Occlusal contact area was measured using Dental Prescale II (GC Japan, Tokyo, Japan). Briefly, the patients firstly practiced the biting of prescale film, and then, the prescale film was firmly bitten three times. The occlusal contact area was calculated by reading it with scanner. Mean value of three measurements was used for experiments. Prior to the experiments, we calibrated the measurements and confirmed intraclass correlation coefficient (ICC) values after two weeks. ICC value was 0.86, which indicated excellent measurement consistency.

### 2.5. BBF Measurement

BBF was measured using functional NIRS (fNIRS; OEG-16 device; Spectratech, Tokyo, Japan), which measures oxygenated hemoglobin (oxy-Hb) in the prefrontal cortex via 16 channels. The details of the BBF measurements have been previously described elsewhere [4,5]. Among the 16 channels, channels 1 to 4 and 13 to 16, containing the inferior frontal gyrus, were analyzed because the inferior frontal gyrus has been reported to play an important role in cognitive function [24,25]. The temporal characteristics of the OEG-16 measurements are defined by a physiological signal bandwidth of 0.76 Hz, with data acquisition performed at 0.65 s intervals.

Prior to the experiments, we calibrated the measurements and confirmed high level of reproducibility in brain blood flow response during chewing, with ICC values exceeding 0.7 even after one week [4]. To ensure consistent fNIRS data capture across trials, we used standard probe placement protocols with conventional headgear. Participants were instructed to maintain a stable head position during measurements to minimize motion artifacts. Additionally, we employed signal processing that included filtering to remove physiological noise and ensured adequate signal quality before proceeding with data analysis. All measurements were conducted in a quiet environment with minimal external stimuli to reduce potential confounding factors.

### 2.6. Task Setting for fNIRS

We used a simple block design consisting of control (resting interval) task and experimental tasks. The experimental tasks were to chew soft CRT paraffin (Ivoclar vivadent, Tokyo, Japan) or hard gummy candy (UHA Mikakuto, Osaka, Japan) on their preferred side. Each task was performed for approximately 1 min with participants chewing at their natural rhythm (approximately 1 to 2 Hz, 30 to 70 cycles). To ensure reasonable consistency in chewing patterns, the investigator monitored all sessions, and participants were instructed to chew on their preferred side throughout all trials. While individual chewing rates were not strictly standardized, participants were asked to maintain a comfortable, consistent pace during each task to minimize variability within individual measurements. The task was terminated if the maximum oxy-Hb value was obtained earlier as monitored in real time. This approach ensured standardization of the masticatory effort while allowing for individual variations in hemodynamic response times. After each task, control task (a resting interval of at least 30 s) was implemented to allow oxy-Hb levels to return to baseline. The return to baseline was confirmed by monitoring the oxy-Hb signal before proceeding to the next task.

### 2.7. fNIRS Data Analyses

The details of fNIRS data analyses have been described elsewhere [4,5]. Briefly, the maximum values of oxy-Hb in each channel during each task and time-to-peak BBF response were analyzed in this study. The maximum oxy-HB values were calculated for each task and then compared between the tasks and the groups. The time-to-peak BBF response was calculated as the duration between task onset and the point at which the oxy-Hb value reached its maximum, and these were compared between the tasks and groups. For quantifying the maximum oxy-Hb values, we used the highest value among the maximum values from channels 1–4 and channels 13–16 as the representative value for each participant. This approach differs from our previous study [4], where we calculated the median of maximum values across channels. The highest value method was selected for this study, as it better captures the peak hemodynamic response necessary for accurate temporal analysis, particularly for determining precise time-to-peak measurements. This analytical approach, supported by Pinti et al. [20], allows for more sensitive detection of localized peak responses while potentially introducing greater individual variation compared to median-based methods.

### 2.8. Statistical Analysis

Statistical analyses were performed using both conventional non-parametric testing (Mann–Whitney U test) and bootstrap analysis to ensure robust interpretation of our findings. The Mann–Whitney U test was chosen due to the non-normal distribution of our data and its robustness against outliers. Additionally, we employed bootstrap analysis with 10,000 resamples to provide more detailed insights into the reliability and stability of our findings, particularly given our moderate sample size. While the Mann–Whitney U test provided traditional hypothesis testing with a significance threshold of *p* < 0.05, the bootstrap analysis offered complementary information through confidence intervals, allowing us to assess the precision and consistency of observed differences between groups. A statistically meaningful difference between groups was confirmed when the 95% confidence interval of the difference between medians did not contain zero, indicating a consistent directional effect. This dual analytical approach provided a more comprehensive understanding of the data, with bootstrap analysis being particularly valuable in cases where traditional *p*-values were near the significance threshold. Specifically, bootstrap analysis strengthens our statistical inference by providing distribution-free estimates of confidence intervals, allowing for more reliable assessment of effect sizes in our moderate sample size study [26]. Correlation analysis between the factors was performed using Spearman’s rank correlation analysis.

All statistical analyses were performed with SPSS Statistics (version 27.0, IBM, Tokyo, Japan).

## 3. Results

### 3.1. Characteristics of NORM and MP Groups

The mean (SD) ages of the NORM and MP groups were 22.7 (4.4) and 21.9 (6.0) years, respectively, which were similar (*p* = 0.292, *t*-test). There was no difference in the ratio of men to women between the groups (*p* = 0.732, Fisher exact test). Cephalometric analysis of the MP group revealed that there was a significant difference to the Japanese normal standard in ANB angle (−2.3 ± 1.6 vs. 3.1 ± 2.5, respectively; *p* < 0.001, *t*-test) [27] but not in Frankfort mandibular angle (FMA) (*p* = 0.843, *t*-test). These results signify that the MP group exhibited marked anteroposterior skeletal disharmony between the maxilla and mandible.

Then, we compared the occlusal contact area and found that MP exhibited quite a small occlusal contact area as compared to that in NORM (Figure 1).

### 3.2. Intragroup Comparison of Maximum Oxy-Hb Between Test Foods

When comparing the BBF responses between paraffin and hard gummy candy within each group, no significant differences were observed in either the NORM group (Figure 2A) or the MP group (Figure 2B; *p* > 0.05). The statistical analysis summary for maximum oxy-Hb is described in Table 1.

These results indicate that the hardness of the test food did not significantly affect the magnitude of the BBF response within each group.

### 3.3. Intergroup Comparison of Maximum Oxy-Hb for Paraffin and Hard Gummy Candy Mastication

During paraffin mastication, patients in the MP group showed a tendency toward lower BBF responses compared to individuals with normal occlusion, although this difference did not reach statistical significance (Figure 2C; *p* > 0.05). The median values were 0.84 for the NORM group and 0.67 for the MP group.

In contrast to paraffin mastication, the BBF response during hard gummy candy mastication revealed a significant difference between groups (Figure 2D; *p* < 0.05). Patients with MP demonstrated significantly lower BBF responses (median: 0.45) compared to those with normal occlusion (median: 1.16). This finding suggests that the masticatory dysfunction associated with MP more prominently affects BBF responses when masticating firmer materials.

### 3.4. Hard Gummy Candy/Paraffin Ratio Comparison of Maximum Oxy-Hb

Comparison of the hard gummy candy/paraffin ratio of the maximum oxy-Hb values between groups revealed a significant difference. The NORM group demonstrated a higher hard gummy candy/paraffin ratio (median: 1.23) compared to the MP group (median: 0.74, *p* < 0.05; Figure 3), suggesting that individuals with normal occlusion exhibit a proportionally enhanced BBF response to increased food hardness, whereas those with MP demonstrate an attenuated cerebrovascular response to harder food materials.

### 3.5. Intragroup Comparison of Time-to-Peak BBF Response Between Test Foods

Within the NORM group, the time taken to reach peak BBF for paraffin mastication and hard gummy candy mastication was 28.75 (range: 12.00–56.25) and 56.50 (range: 22.62–76.38), respectively (Figure 4A). The Mann–Whitney U analysis revealed no significant difference between the tasks (*p* = 0.052). On the other hand, the bootstrap analysis produced 95% confidence intervals containing only negative values, suggesting that the hard gummy candy task consistently demonstrated higher values compared to the paraffin task (Table 2).

Additionally, the MP group demonstrated a significantly prolonged time-to-peak BBF response during hard gummy candy mastication compared to paraffin mastication (Figure 4B and Table 2).

The median value for time-to-peak BBF during paraffin mastication was approximately 20 s in both groups, while hard gummy candy mastication required approximately 40 s. The statistical analysis summary for time-to-peak BBF is described in Table 2.

### 3.6. Intergroup Comparison of Time-to-Peak BBF Response for Paraffin and Hard Gummy Candy Mastication

During paraffin mastication, no significant difference was observed in the time-to-peak BBF between the NORM and MP groups (Figure 4C; *p* > 0.05). The median values were comparable, with both groups requiring approximately 20 s to reach peak BBF response.

Similarly, during hard gummy candy mastication, there was no significant difference in the time-to-peak BBF between the NORM and MP groups (Figure 4D; *p* > 0.05). Both groups exhibited prolonged time-to-peak values of approximately 40 s when masticating the firmer gummy material.

### 3.7. Hard Gummy Candy/Paraffin Ratio Comparison of Time-to-Peak BBF Response

The analysis of the hard gummy candy/paraffin ratio of the time-to-peak response revealed no significant difference between the NORM and MP groups (Figure 5; *p* > 0.05). This finding suggests that both groups demonstrate similar proportional increases in time-to-peak BBF when transitioning from softer to harder food materials, indicating that the temporal adaptation to increased food hardness is preserved in patients with MP despite their altered masticatory function.

### 3.8. Correlation Analysis of the Factors Against BBF

Finally, we performed a correlation analysis of the occlusal contact area to the BBF factors (Table 3). The occlusal contact area exhibited significant positive and moderate correlation to the maximum oxy-Hb value both in paraffin (ρ = 0.450, *p* = 0.046) and hard gummy candy tasks (ρ = 0.520, *p* = 0.019). On the other hand, the occlusal contact area exhibited no significant correlation to time-to-peak BBF in both tasks (paraffin: ρ = 0.263, *p* = 0.262; hard gummy candy: ρ = −0.363, *p* = 0.116). These results suggest that the extent of the occlusal contact area influences the magnitude of BBF but not the temporal dynamics of BBF increase during mastication.

The correlation analysis of ANB to BBF revealed no statistically significant correlation both for the maximum oxy-Hb value and time-to-peak in paraffin and hard gummy candy tasks (Table 4). These results suggest that the extent of jaw deformity judged by ANB gives marginal impact on the retardation of chewing-induced BBF.

## 4. Discussion

Our study investigated two specific hypotheses: (1) that food hardness affects both the magnitude and temporal patterns of BBF during mastication and (2) that malocclusion is associated with alterations in these hemodynamic responses. Our findings partially supported these hypotheses, revealing a dissociation between magnitude and temporal effects. Food hardness significantly extended the time-to-peak BBF in both groups (approximately doubling from ~20 s to ~40 s) while not significantly affecting the maximum oxy-Hb values within either group. Patients with MP showed significantly attenuated BBF magnitudes during hard food mastication compared to the controls, but both groups exhibited similar temporal adaptation patterns. Additionally, the occlusal contact area exhibited positive correlation to BBF magnitude but not to temporal hemodynamic responses. These findings indicate a differential effect of MP on BBF responses: while the magnitude of the response is compromised, the temporal orchestration remains intact.

Further analysis of our intergroup comparisons revealed important differences in the relationship between food hardness and BBF. While MP patients showed only minor differences in BBF response during paraffin mastication compared to the control subjects, the contrast became significantly more pronounced during hard gummy candy mastication. This finding builds upon our previous work [4] and suggests that challenging masticatory conditions amplify the cerebrovascular differences between normal and pathological occlusion. The hard gummy candy/paraffin ratio analysis further reinforced this observation, showing that individuals with normal occlusion exhibit proportionally enhanced BBF responses to increased food hardness, whereas those with MP demonstrate an attenuated cerebrovascular response. These findings add a new dimension to the previous findings by Hasegawa et al. [16] and extend our understanding of BBF responses in patients with jaw deformities. Our work aligns with studies by Kimoto et al. [21], who documented altered neuromuscular patterns in patients with jaw deformities. Our methodology offers important advantages for examining these temporal relationships, as NIRS provides superior temporal resolution compared to fMRI for monitoring rapid hemodynamic changes. Work by Piper et al. [28] has validated this temporal advantage of NIRS, while Sato et al. [19] have demonstrated its particular utility in studying cognitive function, supporting our methodological approach.

The correlation analysis revealed significant associations between the occlusal contact area and the magnitude of masticatory-induced BBF but not with temporal BBF dynamics. This finding provides important mechanistic insights into the relationship between occlusal function and cerebrovascular responses. The positive correlations between the occlusal contact area and maximum oxy-Hb values during both paraffin and hard gummy candy mastication (ρ = 0.450 and ρ = 0.520, respectively) suggest that the extent of occlusal contact directly influences the magnitude of neural activation and subsequent hemodynamic response. This relationship demonstrates how structural components of masticatory function translate to neurophysiological responses. The occlusal contact area represents a critical structural parameter that directly affects masticatory efficiency and force distribution. Previous research by Iwase et al. [3] has demonstrated that patients with MP exhibit reduced masticatory efficiency and altered muscle recruitment patterns. Our finding that the occlusal contact area correlates with BBF magnitude but not temporal dynamics suggests that while structural occlusal parameters influence the amplitude of cerebrovascular responses, they may not significantly affect the temporal orchestration of these responses. This dissociation between structural influences on magnitude versus timing aligns with research by Hirano et al. [29], who found that mastication-induced neural activation involves complex sensorimotor integration that may be differentially affected by structural and functional parameters. Furthermore, the relationship between reduced occlusal contact area in MP patients and their attenuated BBF responses during hard food mastication highlights the functional consequences of altered occlusal relationships. In clinical contexts, this suggests that orthognathic interventions to normalize occlusal contacts may potentially enhance cerebrovascular responses during mastication, with possible implications for the cognitive benefits associated with efficient mastication [11,26].

From a clinical perspective, these findings indicate that occlusal rehabilitation strategies focusing on optimizing the contact area and distribution may improve the magnitude of masticatory-induced BBF in patients with craniofacial deformities, potentially enhancing the neurocognitive benefits associated with efficient mastication. Our focus on the total occlusal contact area rather than detailed contact distribution patterns builds upon our previous work on anterior open bite [5], where we demonstrated the relationship between specific contact pattern alterations and BBF responses. Anterior open-bite cases exhibited no occlusal contact in the anterior region of dental arch, while mandibular prognathism cases have some occlusal contact in the anterior region of dental arch. This distinction may contribute to the different patterns of BBF response observed across malocclusions. Our current study primarily focused on occlusal contact area; a more comprehensive assessment, including bite force measurements and muscle function evaluation, would provide additional insights. In our previous research, we demonstrated that MP patients exhibited significantly lower electromyographic (EMG) values in both masseter and temporal muscles compared to subjects with normal occlusion [4], which aligns with the reduced BBF response observed in the current study. The difference in occlusal contact pattern between open bite and MP might also influence the severity of chewing-induced BBF reduction between these malocclusions. We found that the total occlusal contact area showed a moderate positive correlation with the maximum oxy-Hb values during both tasks, suggesting that the overall extent of occlusal contacts plays a significant role in determining the magnitude of masticatory-induced cerebrovascular responses. This approach allows for comparison across different types of malocclusions (anterior open bite in our previous work and mandibular prognathism in the current study), contributing to a more comprehensive understanding of how various craniofacial deformities affect masticatory function and associated cerebrovascular dynamics.

Our observation that temporal adaptation patterns to increased food hardness are preserved in MP patients despite attenuated BBF magnitude represents a significant contribution to our understanding of neurovascular coupling in craniofacial deformities. Previous investigations of masticatory function in MP patients have documented various functional impairments but have focused almost exclusively on magnitude-based measurements, such as reduced bite force, altered muscle activity patterns, and decreased masticatory efficiency [3]. Our study is the first to distinguish between the magnitude and timing aspects of cerebrovascular responses during mastication in these patients. This distinction reveals a nuanced picture of neurofunctional adaptation in MP: while the amplitude of BBF response is compromised (particularly during hard food mastication), the temporal orchestration of this response remains intact. This dissociation between magnitude and timing parameters suggests differential effects of craniofacial morphology on distinct aspects of neurovascular coupling mechanisms. Indeed, the extent of jaw deformity judged by ANB gave no correlation to BBF. The preserved temporal dynamics indicate that the basic neural circuitry governing the timing of cerebrovascular responses to increased masticatory demand remains functional in MP patients, even as the magnitude of blood flow mobilization is reduced. This finding challenges simplistic views of masticatory dysfunction in craniofacial deformities and suggests more complex, selective alterations in neurovascular regulation that have not been previously documented.

The different patterns observed in our group comparisons compared to our previous work [4] may be partially explained by our methodological approach to quantifying oxy-Hb values. As detailed in our Methods section, the present study employed the highest value among maximum values from multiple channels, while our previous study used median values. This discrepancy suggests that using median values might better reflect the overall hemodynamic response across multiple channels, while the highest value approach might be more sensitive to localized peak responses, as demonstrated by Ye et al. [30]. This approach aligns with analytical recommendations by Pinti et al. [20] for processing fNIRS data when examining temporal characteristics. While potentially introducing more individual variation, this approach was particularly appropriate for our current focus on temporal dynamics. These methodological considerations highlight the importance of careful selection of data processing methods in fNIRS studies, as they can significantly influence the interpretation of masticatory-induced hemodynamic responses. Future studies comparing these different analytical approaches within the same dataset would be valuable for establishing optimal processing methods for masticatory fNIRS data.

Several limitations should be acknowledged when interpreting the findings of this study. First, the relatively moderate sample size (NORM: *n* = 26, MP: *n* = 20) may limit the statistical power to detect subtle group differences, particularly in temporal patterns of BBF responses. A post hoc power analysis indicated that our achieved power was 0.68, suggesting that a larger sample might provide more robust evidence. Recent systematic reviews by Zafar et al. [31] of masticatory function studies emphasize the importance of an adequate sample sizing for detecting subtle neurovascular responses. Second, our study focused exclusively on young Japanese adults (mean age: 22.7 ± 4.4 years for NORM group; 21.9 ± 6.0 years for MP group), which reflects the typical demographic seeking surgical orthodontic treatment for mandibular prognathism. While age-matching our control group to the MP group strengthened our internal validity by eliminating age as a confounding variable, this approach limits the generalizability of our findings to middle-aged and older populations and other ethnicities. This consideration is particularly relevant given that masticatory function and cerebrovascular regulation both undergo significant changes with aging. Previous research has suggested that mastication may have different cognitive and hemodynamic effects in older populations [11,26], highlighting the importance of age-specific investigations. Third, the fNIRS technique, while offering excellent temporal resolution, is limited to measuring hemodynamic responses in cortical regions and cannot capture activity in deeper brain structures that may be involved in masticatory processing. Fourth, individual variations in masticatory habits, diet preferences, and neuromuscular coordination were not systematically controlled in our analysis. Specifically, the limited assessment of masticatory function represents a notable limitation, as we measured only the occlusal contact area without a comprehensive evaluation of bite force or masticatory muscle function. Finally, the cross-sectional design of our study precludes establishing causal relationships between MP and altered BBF patterns.

Statistical analyses of the time-to-peak BBF revealed interesting patterns across different testing methodologies. While the Mann–Whitney U test for the NORM group yielded a *p*-value of 0.052, marginally above the conventional significance threshold of 0.05, the bootstrap analysis provided additional insights. Specifically, the 95% confidence intervals for the difference in medians (−44.38 and −1.62) were both negative, suggesting a consistent tendency for higher values during the hard gummy candy task compared to the paraffin task. A similar pattern was observed in the MP group, where despite differences in *p*-values between the Mann–Whitney U test (*p* = 0.250) and bootstrap analysis (*p* = 0.008), the 95% confidence intervals (−49.00 and −7.50) again contained only negative values. These findings, particularly the non-overlapping confidence intervals with zero, provide compelling evidence for a systematic difference between tasks, even when traditional significance testing approaches yield borderline results. This underscores the value of employing multiple statistical approaches to fully characterize treatment effects, especially in cases where conventional significance testing may not capture the full extent of the underlying patterns in the data. Further confirmation is necessary with enforcing the statistical power by increasing the number of subjects.

The temporal characteristics of BBF responses, particularly time-to-peak BBF, provide important insights into the neurovascular coupling mechanisms during mastication. Our observation that soft food induces more rapid BBF peaks (~20 s) compared to hard food (~40 s) suggests more efficient neurovascular coupling with less demanding masticatory tasks. This finding aligns with previous research by Hasegawa et al. [16], who demonstrated that excessive masticatory load could actually lead to decreased cerebral blood flow. The shorter time-to-peak BBF observed with softer food might reflect optimal neural activation patterns, as supported by Sato et al. [19], who showed that rapid hemodynamic responses are associated with more efficient neural processing. This interpretation is further supported by research demonstrating that efficient neurovascular coupling typically involves prompt blood flow responses to neural activation [29]. The observed temporal differences in reaching peak blood flow between soft and hard materials may reflect the increased neuromuscular demands associated with harder food mastication. Work by Ohkubo et al. [22] has demonstrated that masticatory effort correlates with specific patterns of cortical activation, supporting our findings of extended temporal responses to harder materials. Furthermore, Lin et al. [32] has shown that masticatory force modulation involves complex neurofeedback mechanisms, which may explain the observed differences in temporal dynamics. The preservation of temporal patterns in MP patients, despite a reduced magnitude of response, suggests intact basic neurovascular coupling mechanisms. The temporal characteristics of masticatory-induced BBF changes revealed in this study have several important clinical implications. Research by Ko et al. [33] has demonstrated that understanding masticatory function is crucial for post-orthognathic surgery rehabilitation. The extended time required for peak blood flow during hard food mastication suggests the need for graduated dietary progression during rehabilitation. Furthermore, Chen et al. [34] has shown that mastication plays a crucial role in cognitive function and BBF regulation, emphasizing the importance of our temporal findings in broader neurophysiological contexts. Consistently, Ohkubo et al. [22] also reports that masticatory function is important for maintaining cognitive function, and chewing during exposure to stress might be a useful method of coping with stress.

The present findings have several specific clinical implications that directly emerge from our data:
(1)Progressive Dietary Protocol: Our observation that harder foods require approximately twice the time to reach peak BBF (40 s vs. 20 s) provides concrete guidance for post-orthognathic surgery rehabilitation. Patients should be allowed sufficient time for masticating harder foods to achieve maximum cerebrovascular activation.(2)Occlusal Contact Optimization: The significant positive correlation between the occlusal contact area and maximum oxy-Hb values (ρ = 0.450 for paraffin; ρ = 0.520 for hard gummy candy) suggests that surgical and orthodontic interventions should prioritize maximizing the functional occlusal contact area to enhance cerebrovascular response during mastication.(3)Preserved Temporal Adaptation: Since MP patients maintain normal temporal adaptation patterns despite reduced BBF magnitude, rehabilitation protocols should focus on gradually increasing food hardness to improve BBF response magnitude while leveraging intact temporal adaptation mechanisms.(4)Extended Mastication for Cognitive Benefits: Given the established link between mastication and cognitive function [11,26], MP patients might benefit from customized mastication exercises with extended chewing periods to compensate for their reduced BBF magnitude, potentially optimizing cognitive benefits despite their altered masticatory function.(5)Monitoring Timeframes: Clinicians should consider that MP patients require similar timeframes as those with normal occlusion to reach peak BBF (approximately 40 s for harder foods), which provides important parameters for designing functional assessments following orthognathic surgery.


These data-driven recommendations transform our findings on temporal BBF characteristics into practical guidance for clinical rehabilitation strategies following orthognathic surgery.

Several promising directions for future research emerge from this study. First, a longitudinal investigation of BBF changes before and after orthognathic surgery would provide valuable insights into the neuroplastic adaptation following correction of skeletal discrepancies. Such studies could employ repeated fNIRS measurements at different timepoints during post-surgical rehabilitation to track changes in both the magnitude and temporal characteristics of masticatory-induced BBF. Second, a multimodal neuroimaging approach that combines the complementary strengths of fNIRS and functional magnetic resonance imaging (fMRI) would provide a more comprehensive understanding of masticatory-induced neural activity. While fMRI offers superior spatial resolution allowing for the precise localization of activated brain regions during mastication, it is limited by poor temporal resolution (typically 2–3 s). In contrast, our fNIRS approach provides excellent temporal resolution (sub-second level) that is critical for characterizing the temporal dynamics of BBF responses, as demonstrated in this study. Integration with emerging technologies, such as simultaneous NIRS-electroencephalography recordings, as demonstrated by Noah et al. [35], would further enhance our understanding of the neural mechanisms underlying masticatory function. Combining these techniques in future research would enable simultaneous assessment of both the precise spatial distribution of masticatory-induced neural activation (via fMRI) and the detailed temporal characteristics of these responses (via fNIRS). Third, future studies should investigate the relationship between distribution patterns of occlusal contact and BBF dynamics to determine whether specific occlusal rehabilitation strategies might optimize neurovascular responses during mastication. Future studies should incorporate a comprehensive assessment of masticatory function, including bite force measurements, electromyographic analysis of masticatory muscles, and masticatory efficiency evaluations alongside BBF measurements. This integrated approach would provide more thorough insights into the relationship between masticatory function parameters and cerebrovascular responses during mastication, potentially revealing specific functional markers that correlate with temporal BBF dynamics. Our findings demonstrate that the timing to achieve maximum brain blood flow during mastication remains consistent between individuals with normal occlusion and those with mandibular prognathism. This temporal stability provides a foundation for standardized clinical assessment protocols. By measuring brain blood flow at predetermined intervals following mastication onset—approximately 20 s for soft materials and 40 s for harder foods—clinicians could systematically quantify the magnitude of cerebrovascular impairment attributable to malocclusion. This standardized approach would enable an objective assessment of the systemic neurological effects of craniofacial deformities, providing valuable biomarkers for treatment planning and outcome evaluation beyond traditional measures of masticatory efficiency. Such protocols could facilitate evidence-based clinical decision-making regarding the broader health implications of malocclusion and the monitoring of therapeutic interventions. As to the use of our results in rehabilitation strategies, the temporal characteristics revealed in this study suggest specific post-surgical protocols for orthognathic patients. The consistent doubling of time-to-peak brain blood flow for harder materials indicates that dietary progression should incorporate extended mastication periods, with patients maintaining mastication of firmer foods for at least 40–50 s rather than typical 20–30 s periods. The preserved temporal adaptation mechanisms in mandibular prognathism patients suggest that rehabilitation should emphasize progressive hardness training rather than avoidance of challenging foods, potentially enhancing brain blood flow response magnitude while leveraging intact temporal coordination mechanisms. Regarding surgical and post-operative intervention strategies, the significant positive correlation between the occlusal contact area and brain blood flow magnitude provides direct clinical guidance. Surgeons should prioritize achieving the maximum functional occlusal contact area during orthognathic procedures, as incremental improvements in the contact area may translate to enhanced cerebrovascular responses during mastication. Post-surgical orthodontic finishing should focus specifically on optimizing posterior occlusal contacts, which contribute substantially to total contact area and subsequent neurovascular benefits during mastication. Finally, exploration of potential correlations between BBF parameters and patient-reported outcomes, such as masticatory satisfaction and quality of life measures, would enhance our understanding of the clinical significance of these hemodynamic changes and provide deeper insights into the neurofunctional adaptations associated with craniofacial deformities.

## 5. Conclusions

Our cross-sectional observational study examining the temporal characteristics of masticatory-induced brain blood flow revealed two primary findings:

First, food hardness significantly affects the temporal dynamics of BBF during mastication, requiring approximately twice the time to reach peak levels for harder foods compared to softer materials regardless of occlusal condition. This consistent temporal pattern was observed in both normal occlusion and mandibular prognathism groups.

Second, while patients with mandibular prognathism showed significantly attenuated BBF magnitudes during mastication, particularly with harder foods, their temporal adaptation patterns remained intact. This dissociation between magnitude and timing suggests that the basic neurovascular coupling mechanisms governing temporal hemodynamic responses are preserved despite altered masticatory function in MP.

Additionally, the occlusal contact area showed significant positive correlation with BBF magnitude, suggesting that structural occlusal parameters may partially mediate the observed BBF alterations in MP patients.

These findings provide new insights into masticatory-induced cerebrovascular dynamics and have direct clinical implications for rehabilitation strategies following orthognathic surgery. The preserved temporal orchestration of neurovascular responses, despite reduced BBF magnitude, suggests that rehabilitation protocols should focus on gradually increasing food hardness while allowing for sufficient time for peak cerebrovascular activation, potentially optimizing both masticatory function and cognitive benefits associated with efficient mastication.

## Figures and Tables

**Figure 1 dentistry-13-00250-f001:**
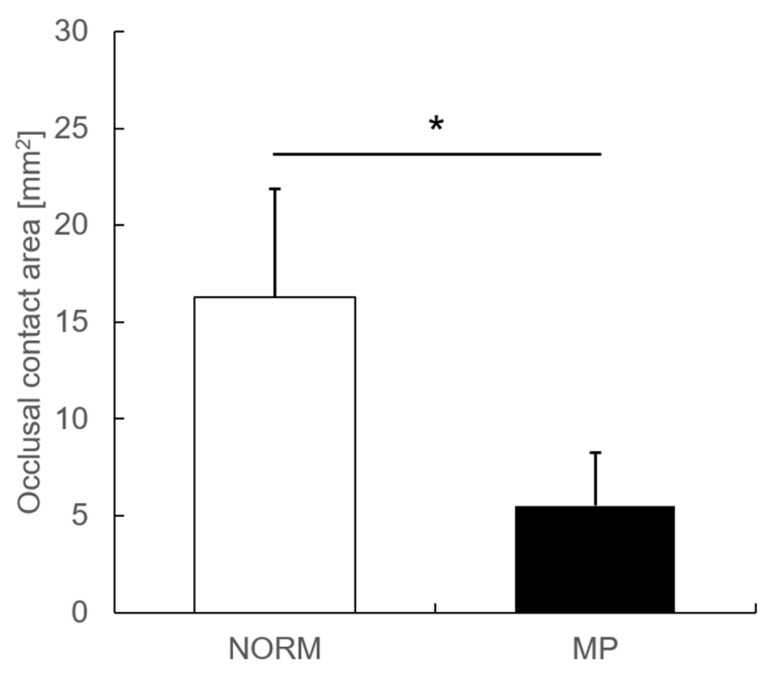
Comparison of occlusal contact area between NORM and MP groups. Occlusal contact area of both NORM and MP groups were measured. NORM, normal occlusion; MP, mandibular prognathism; * significant difference (*p* < 0.05).

**Figure 2 dentistry-13-00250-f002:**
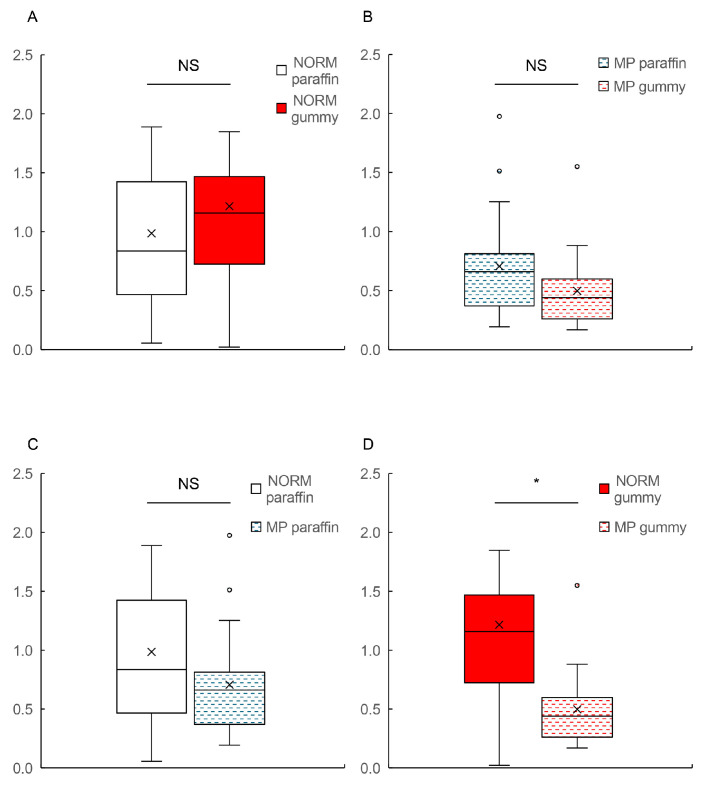
A comparison of the maximum oxy-Hb values between the NORM and MP groups with paraffin and hard gummy candy mastication tasks. (**A**) A comparison of the maximum oxy-Hb values between paraffin (white box) and hard gummy candy (red box) mastication tasks in the NORM group. (**B**) A comparison of the maximum oxy-Hb values between paraffin (blue wave-patterned box) and hard gummy candy (red wave-patterned box) mastication tasks in the MP group. (**C**) A comparative analysis of the maximum oxy-Hb values during paraffin chewing between the NORM (white box) and MP (blue wave-patterned box) groups. (**D**) A comparative analysis of the maximum oxy-Hb values during hard gummy candy chewing between the NORM (red box) and MP (red wave-patterned box) groups. Box plots represent median (horizontal line), mean (×), 25th and 75th percentiles (boxes), and minimum and maximum values (whiskers). Outliers are shown as individual points (○). The Y-axis indicates the oxy-Hb value [a.u.]. oxy-HB, oxygenated hemoglobin; NORM, normal occlusion; MP, mandibular prognathism; NS, not significant. * significant difference.

**Figure 3 dentistry-13-00250-f003:**
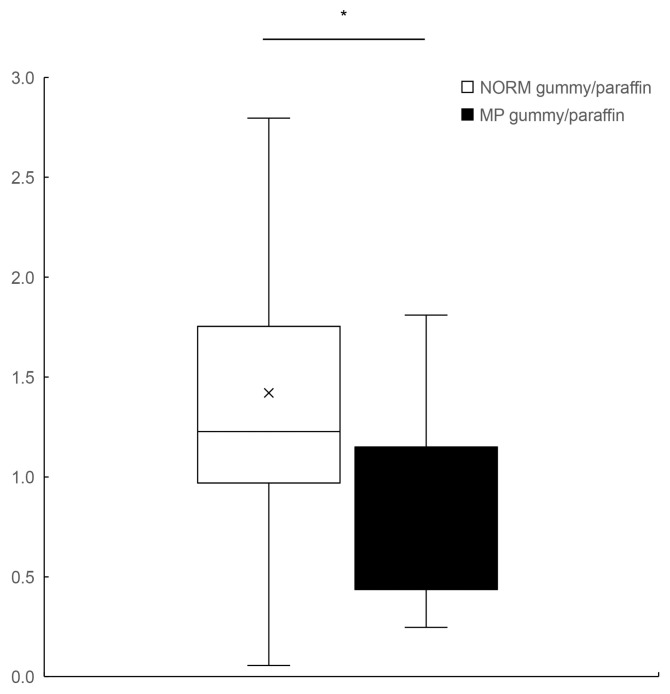
A comparison of the hard gummy candy/paraffin ratio of the maximum oxy-Hb values between the NORM (white box) and MP (black box) groups. Box plots represent median (horizontal line), mean (×), 25th and 75th percentiles (boxes), and minimum and maximum values (whiskers). Outliers are shown as individual points (○). oxy-HB, oxygenated hemoglobin; NORM, normal occlusion; MP, mandibular prognathism. * significant difference between the groups.

**Figure 4 dentistry-13-00250-f004:**
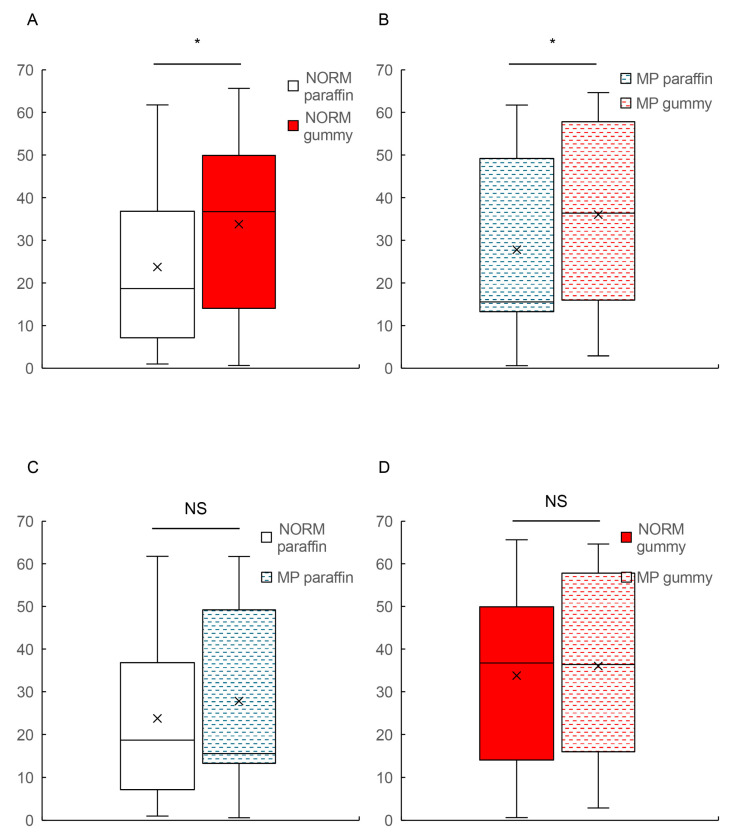
A comparison of time-to-peak BBF between the NORM and MP groups during paraffin and hard gummy candy mastication tasks. (**A**) A comparison of time-to-peak BBF between paraffin (white box) and hard gummy candy (red box) mastication tasks in the NORM group. (**B**) A comparison of time-to-peak BBF between paraffin (blue wave-patterned box) and hard gummy candy (red wave-patterned box) mastication tasks in the MP group. (**C**) A comparative analysis of time-to-peak BBF during paraffin chewing between the NORM (white box) and MP (blue wave-patterned box) groups. (**D**) A comparative analysis of time-to-peak BBF during hard gummy candy chewing between the NORM (red box) and MP (red wave-patterned box) groups. Box plots represent median (horizontal line), mean (×), 25th and 75th percentiles (boxes), and minimum and maximum values (whiskers). Outliers are shown as individual points (○). Y-axis indicates the time-to-peak [sec.]. BBF, brain blood flow; NORM, normal occlusion; MP, mandibular prognathism; NS, not significant. * significant difference.

**Figure 5 dentistry-13-00250-f005:**
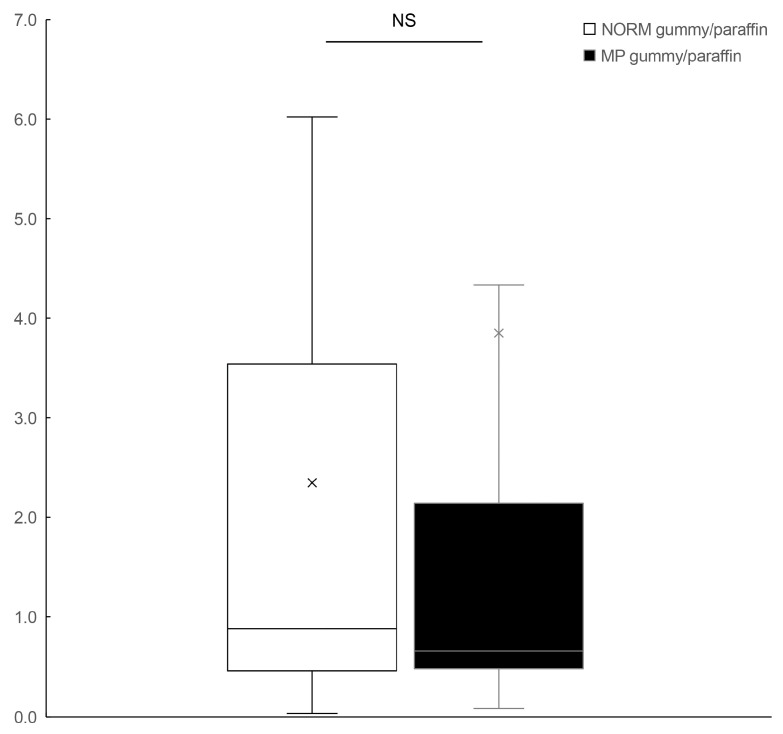
A comparison of the hard gummy candy/paraffin ratio of time-to-peak BBF between the NORM (white box) and MP (black box) groups. Box plots represent median (horizontal line), mean (×), 25th and 75th percentiles (boxes), and minimum and maximum values (whiskers). Outliers are shown as individual points (○). BBF, brain blood flow; NORM, normal occlusion; MP, mandibular prognathism; NS, not significant. A comparison of time-to-peak BBF between the NORM and MP groups during paraffin and hard gummy candy mastication tasks.

**Table 1 dentistry-13-00250-t001:** Summary of statistical analysis for maximum Oxy-Hb.

Comparison	MW-U (*p* Value)	Bootstrap (*p* Value)	Bootstrap (95% CI)	Significance
NORM-paraffin VS NORM-gummy	0.227	0.247	−0.81, 0.17	NS
MP-paraffin VS MP-gummy	0.135	0.138	−0.09, 0.64	NS
NORM-paraffin VS MP-paraffin	0.176	0.628	−0.29, 0.47	NS
NORM-gummy VS MP-gummy	0.001	0.004	0.31, 1.06	*
NORM gummy/paraffin VS MP gummy/paraffin	0.006	0.015	0.08, 0.80	*

*p* value of Mann–Whitney U test (MW-U), *p* value of bootstrap analysis, 95% confidence interval (95% CI) of the difference of medians, and significance are indicated. *: statistically significant difference. NS: no significant difference.

**Table 2 dentistry-13-00250-t002:** Summary of statistical analysis for time-to-peak BBF.

Comparison	MW-U (*p* Value)	Bootstrap (*p* Value)	Bootstrap (95% CI)	Significance
NORM-paraffin VS NORM-gummy	0.052	0.071	−44.38, −1.62	*
MP-paraffin VS MP-gummy	0.250	0.008	−49.00, −7.50	*
NORM-paraffin VS MP-paraffin	0.444	0.662	−15.00, 23.50	NS
NORM-gummy VS MP-gummy	0.666	0.960	−19.00, 20.00	NS
NORM gummy/paraffin VS MP gummy/paraffin	0.419	0.518	−0.46, 1.24	NS

*p* value of Mann–Whitney U test (MW-U), *p* value of bootstrap analysis, 95% confidence interval (95% CI) of the difference of medians, and significance are indicated. *: statistically significant difference. NS: no significant difference.

**Table 3 dentistry-13-00250-t003:** Summary of correlation analysis of occlusal contact area to BBF.

Factor		*p* Value	*ρ*
maximum oxy-Hb	paraffin	0.046	0.450 *
	hard gummy candy	0.019	0.520 *
time-to-peak BBF	paraffin	0.262	0.263
	hard gummy candy	0.116	−0.363

*ρ*: Spearman’s rank correlation coefficient, *: statistical significant correlation.

**Table 4 dentistry-13-00250-t004:** Summary of correlation analysis of ANB to BBF.

Factor		*p* Value	*ρ*
maximum oxy-Hb	paraffin	0.80	0.05
	hard gummy candy	0.55	−0.14
time-to-peak BBF	paraffin	0.73	0.08
	hard gummy candy	0.80	−0.06

*ρ*: Spearman’s rank correlation coefficient. There was no statistical significant correlation.

## Data Availability

The data that support the findings of this study are available from the corresponding author upon reasonable request.

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
