# Peer review of "The Occlusal Contact Area Is Associated with the Magnitude but Not Peak Timing of Changes in Chewing-Induced Brain Blood Flow in Mandibular Prognathism"

_dentistry, 2025, doi:10.3390/dj13060250_

Round 1
Reviewer 1 Report (New Reviewer)
Comments and Suggestions for Authors
Peer Review
This study investigates how mandibular prognathism (MP) affects brain blood flow (BBF) during mastication. The authors explore both the magnitude and timing of oxygenated hemoglobin (oxy-Hb) responses using functional near-infrared spectroscopy (fNIRS) in individuals with normal occlusion and those with MP. The work adds to existing research by focusing on the temporal characteristics of BBF, which have been less commonly studied.
General Comments
The manuscript presents a focused research question and offers a relevant methodological approach. The comparison between hard and soft food mastication across occlusal groups is appropriate and generally well explained…However, the manuscript could benefit from more concise structuring and some refinement in how hypotheses and findings are linked in the discussion. Data presentation is clear.
Specific Comments: Introduction:
-The rationale for the study is adequately presented.
Methods:
-The inclusion and exclusion criteria are detailed and appropriate.
-Justification for sample size and statistical methods is sound
-The section describing the use of maximum vs. median oxy-Hb values should be clarified earlier, as it directly affects the analysis logic
-> The exclusion criteria are thorough, but consider briefly noting why only young adults were included, as this affects generalizability.
Results:
-Results are systematically presented with appropriate statistical tests.
-Use of both Mann-Whitney and bootstrap analysis strengthens the robustness of the findings..
Discussion:
-Interpretation of findings is aligned with the results.
-The manuscript occasionally returns to re-explaining the methods and hypotheses, which could be streamlined.
-The impact of occlusal contact area on BBF is an interesting addition, though this point might be better integrated into the broader discussion of masticatory function.
Conclusion:
-Conclusions are supported by the data.
-The potential clinical implications are well noted, especially for post-surgical planning.
- Limitations are acknowledged, though a more direct statement about generalizability would be helpful.
Recommendations for Revision:
- Move and clarify the justification for the “maximum oxy-Hb value” method to the Methods section.
- Simplify and focus the Discussion: avoid re-explaining methods or hypotheses and clearly link each result to interpretation or implication.
- Structure the Discussion more logically, separating technical discussion, limitations, and clinical relevance into distinct paragraphs.
- State the clinical implications more clearly, for exmaple, provide a short bullet list of how these results should influence rehabilitation after orthognathic surgery.
- Add a brief note on generalizability of findings beyond the specific demographic studied (young Japanese adults).
Author Response
Please see the attachment.

Reviewer 2 Report (New Reviewer)
Comments and Suggestions for Authors
This manuscript reported the relationship between mandibular prognathism (MP) and masticatory-induced brain blood flow (BBF), and found that MP attenuates the magnitude of masticatory-induced BBF, particularly during hard food mastication. In view of the previous literature and the authors’ earlier reports, the present data were limited and the results were not sufficient to support the conclusion.
- When evaluating mastication function in normal and MP subjects, only the occlusal contact area was measured. How about the bite forces and the function of muscles.
- The results showed that the occlusal contact area had a significant positive and moderate correlation to the BBF factors. However, the reduced occlusal contact area is not the main characteristic of MP subjects. Therefore, the results and conclusions were not relevant.
Author Response
Please see the attachment.

Reviewer 3 Report (New Reviewer)
Comments and Suggestions for Authors
- Although the introduction extensively reviews the reported literature in this area of research, it could be shortened to eliminate redundant information.
- It would be better to enumerate the inclusion criteria for the two groups as bulleted points.
- Please refer to the previous comment for the exclusion criteria as well.
- Although the outcome measurements were based on electronic instruments, what was their degree of calibration and internal consistency? Provide information about this.
- The results are well described. No comments
- Discussion—Include a paragraph exclusively about the clinical implications of the present outcomes. As a clinician, it would be interesting to note how the study results can be extrapolated when I have a patient with a prognathic mandible and how they would influence treatment outcomes before and after orthognathic surgical correction.
- Similar to the introduction, the discussion can also be trimmed to eliminate redundant information.
- The conclusion should be more objective, based on the study outcomes, and should provide a clear take-home message for the readers. Kindly revise.
Round 2
Reviewer 2 Report (New Reviewer)
Comments and Suggestions for Authors
The result showed that “Occlusal contact area exhibited significant positive correlation with maximum oxy-Hb values, while ANB angle showed no significant correlation with BBF parameters”. I suggest to amend the title as “Occlusal contact area is associated with the magnitude but not peak timing of changes in chewing-induced brain blood flow in Mandibular prognathism”. And the conclusion should be revised accordingly.
Round 3
Reviewer 2 Report (New Reviewer)
Comments and Suggestions for Authors
The title of the article should accurately reflect the primary findings of the experiment and the causal relationships identified. The results indicate that it is the occlusal contact area, rather than MP itself, that is associated with the magnitude—though not with the peak timing—of changes in chewing-induced brain blood flow. I strongly recommend amending the title to: “Occlusal Contact Area Is Associated with Magnitude but Not Peak Timing of Changes in Chewing-Induced Brain Blood Flow in Mandibular Prognathism.”
In conclusion, it is stated that “…is providing new insights for rehabilitation strategies in orthognathic surgery cases.” The authors should provide some examples within the discussion to explain the possible strategies or their clinical implications.
Author Response
Please see the attachment.

This manuscript is a resubmission of an earlier submission. The following is a list of the peer review reports and author responses from that submission.
Round 1
Reviewer 1 Report
Comments and Suggestions for Authors
I would like to express my gratitude to the authors for their efforts on this article. The topic is very interesting as it enhances our understanding and demonstrates the correlation between malocclusion, jaw deformities, and orofacial functions with hemodynamic changes in brain blood flow. Furthermore the methodological approach is innovative and advanced.
The article meets the criteria to be published.
Reviewer 2 Report
Comments and Suggestions for Authors
After a careful review of the manuscript, I would like to present some suggestions and questions that may contribute to the improvement of the manuscript.
Abstract:
- Specify the type of study conducted.
Methods:
- What were the evaluation criteria to rule out temporomandibular dysfunction?
- Indicate whether calibration was performed for data analysis.
- Report if inter-rater reliability was assessed.
- What methods were used to ensure that fNIRS data was captured consistently across trials?
- How was variability in hemodynamic responses across participants addressed? Is there a pattern that suggests that certain individual characteristics (such as age or level of chewing experience) may influence the results?
- Were possible effects of medications participants might be taking or health conditions (such as neurological disorders) that could impact masticatory function considered?
Discussion:
- In the first paragraph, discuss whether the null hypothesis was accepted or rejected.
- Explain the clinical implications of the study and suggest directions for future research in the field.
- What limitations do the authors acknowledge in relation to their study? What other variables that were not considered could influence the results?
Reviewer 3 Report
Comments and Suggestions for Authors
The manuscript addresses the topic described in a rigorous and, in my opinion, original way through a robust statistical analysis that provides solidity to the data presented.
Reviewer 4 Report
Comments and Suggestions for Authors
This study is a interesting research which investigates the mandibular prognathism (MP) affects on brain blood flow (BBF) during chewing.
Through an innovative approach (functional near-infrared spectroscopy (fNIRS)), Authors measured BBF changes, providing insights into the temporal dynamics of hemodynamic responses during mastication.
The study is based on a specific Japanese adult population (which may not be directly applicable to other demographic groups with differing anatomical or dietary characteristics) and meets minimum power requirements, but the sample size (n=46) limits the robustness of the conclusions.
Ethical guidelines are respected and in conclusion key differences in the magnitude of BBF responses between MP and normal occlusion groups are identified, particularly when chewing harder foods. The preservation of temporal patterns despite reduced magnitudes in MP is an interesting and clinically relevant observation.
The study opens new avenues and future research could expand by investigating longitudinal changes in BBF especially post-orthognathic surgery.
Reviewer 5 Report
Comments and Suggestions for Authors
The manuscript does not present significant novelty or contribute new insights to the field. The findings largely reiterate existing knowledge without providing a clear advancement. The authors need to emphasize the unique aspects of their research and its potential impact on the broader scientific community.
Comments on the Quality of English Languagethere are a few minor grammatical errors and awkward sentence constructions